# Latent periodic process inference from single-cell RNA-seq data

Shaoheng Liang [1,2✉], Fang Wang [1], Jincheng Han [3] & Ken Chen [1✉]

The development of a phenotype in a multicellular organism often involves multiple, simultaneously occurring biological processes. Advances in single-cell RNA-sequencing make it possible to infer latent developmental processes from the transcriptomic profiles of cells at various developmental stages. Accurate characterization is challenging however, particularly for periodic processes such as cell cycle. To address this, we develop Cyclum, an autoencoder approach identifying circular trajectories in the gene expression space. Cyclum substantially improves the accuracy and robustness of cell-cycle characterization beyond existing approaches. Applying Cyclum to removing cell-cycle effects substantially improves delineations of cell subpopulations, which is useful for establishing various cell atlases and studying tumor heterogeneity.

[1] Department of Bioinformatics and Computational Biology, The University of Texas MD Anderson Cancer Center, Houston, TX 77030, USA. [2] Department of Computer Science, Rice University, Houston, TX 77005, USA. [3] Department of Cancer Biology, The University of Texas MD Anderson Cancer Center, Houston, TX 77030, USA. ✉email: sliang3@mdanderson.org; kchen3@mdanderson.org

The development of a phenotype in a multicellular organism often involve multiple, simultaneously occurring biological processes such as cell proliferation, differentiation, transition, and cell-to-cell communication[1,2]. The course of development can be influenced by a variety of genetic (e.g., mutations), epigenetic, and environmental factors, which when being abnormally perturbed can result in pathogeneses[3]. Early efforts have been made to reconstruct the temporal ordering of biological samples using bulk data[4–6], although challenges associated with cellular heterogeneity make it difficult to infer accurate time series. Advances in single-cell RNA sequencing (scRNA-seq) enabled large-scale acquisition of single-cell transcriptomic profiles and provided an unprecedented opportunity to uncover latent biological processes that orchestrate dynamic expression of genes in single cells throughout the course of the development[7]. However, it is very challenging to deconvolve these processes from scRNA-seq data accurately. A sufficiently large number of cells across time, lineage, and space need to be sampled in order to capture detailed sub-populational features and reduce technological noise. Tremendous efforts have been made to develop trajectory inference methods from scRNA-seq data. Over 70 methods have been developed since 2014[8], including the widely known Monocle[9] and Wanderlust[10]. These methods represent biological processes in linear, bifurcating, or other graph topologies.

Many developmental processes, such as embryogenesis, organogenesis, and tumorigenesis, are inherently nonlinear[11–15]. For example, cell cycle, a fundamental biological process, is periodic. For human cells, a cycle starts from the G1 phase, goes through S and G2/M, and then returns to G1 within 24 h[2]. This process is orchestrated elegantly by sets of genes (e.g., cyclins and cyclin-dependent kinases) that are turned on and off at relatively precise timings. As a result of such periodicity, the cycling cells at different transcriptomic states form a circular trajectory in high-dimensional gene expression space. The position of a cell alongside the circular trajectory indicates its timing (pseudo-time) in the cell cycle. Although well regulated, the process can be stochastic. For instance, cells can experience different fates (e.g., going into apoptosis or senescence), and the rate of development may fluctuate due to endogenous or exogenous factors[16].

Existing trajectory/pseudo-time inference methods are not optimal for representing such nonlinear periodicity[8]. Those based on linear representations, such as principal component analysis (PCA), cannot accurately represent circular timings or infer effect sizes. The cell cycle regression approaches implemented in scLVM[17], Seurat[18], and ccRemover[19] are based on linear representations generated from user-defined gene sets, which may be biased or incomprehensive, particularly in cancer cells with aberrant cell cycle. scLVM[17] infers latent factors using a linear Gaussian process latent variable model (GPLVM) and identifies which of these factors corresponds to cell cycle by examining correlation with a set of cell-cycle markers. f-scLVM (also known as Slalom)[20] further acquires the ability of refining known gene sets and discovering new ones. Other linear factorization approaches such as non-negative matrix factorization (NMF) and independent component analysis (ICA) have also been explored[21–23]. Although these methods benefit biological discoveries in various ways, they cannot effectively model multi-stage, nonlinear biological processes such as cell cycle. Cyclone[24] uses PCA and relative expression of gene pairs to predict cell-cycle phases, which appears to perform better than traditional machine learning methods, such as random forest, logistic regression, and support vector machine (SVM). A recent method reCAT[25] reconstructs cell-cycle pseudo-time using a Gaussian mixture model (GMM) to cluster single cells into groups and a quasi-optimal traveling salesman path (TSP) solver to order them. The resulting pseudo-time is expressed in consecutive integers indicating the order of cells, instead of continuous real-number timings. Neither Cyclone nor reCAT can be applied to remove cell-cycle effects from the expression data. In addition, Oscope[26] checks pairs of genes to identify circular patterns, which has a high computational burden. Cyclops[6] models circadian rhythm using an autoencoder approach, but employs square root and division in the neural network, which complicates the optimization.

To address these limitations, we develop an ab initio inference method, namely Cyclum, which employs a distinct sinusoidal autoencoder to capture the circular trajectory in high-dimensional gene expression space, formed by single cells sampled from various stages of a periodic process. Conceptually, our approach identifies an optimal (least square) embedding of cells in a circular space. It effectively unfolds the circular manifold onto a linear space to obtain precise pseudo-time.

## Results

**Overview of Cyclum**. In a nutshell, the Cyclum program (Fig. 1) analyzes a cell-gene expression matrix using an autoencoder technique (see Methods and Supplementary Fig. 1), which projects the cells onto a nonlinear periodic trajectory, where the pseudo-times of the cells in a periodic process can be more accurately determined than with linear approaches, such as PCA. Cyclum can be used to identify genes associated with the periodic process, based on the degrees of match between the kinetics of gene expressions and the inferred periodicity. Additionally, this program can treat the inferred periodic process as a confounder and deconvolve its effects from scRNA-seq data. Using Cyclum in this way can result in enhanced delineations of cell subpopulations segregated by lineages or phenotypes.

**Accuracy of Cyclum for cell-cycle characterization**. We compared Cyclum's performance for characterizing cell cycles with Slalom, Cyclone, reCAT, Oscope, Cyclops and PCA. Four datasets were used (Table 1). The first dataset was obtained from a set of mouse embryonic stem cells (mESC) using SMARTer kit and Illumina HiSeq 2000 sequencing technology[17]. The other three datasets were obtained using nanoString nCounter technology from the bone metastasis of a prostate adenocarcinoma (PC3), the pleural effusion metastasis of a breast adenocarcinoma (MB), and the lymphoblast node of a lymphoma (H9), respectively[27]. Each of these datasets has 200–400 cells. Flow sorting with Hoechst staining was performed on the same set of cells, classifying the cells into three stages, G0/G1, S, and G2/M, based on their DNA mass.

We ran each algorithm on each dataset and classified cells into various cell-cycle stages (e.g., Fig. 2a, Supplementary Fig. 2, and Supplementary Note 1). Discretization of the continuous Cyclum and reCAT results was accomplished using a three-component Gaussian mixture model. We then calculated the fraction of cells that were correctly classified by comparing the predicted cell-cycle labels with those obtained from the flow-sorting. As shown (Fig. 2b), for all four datasets Cyclum outperformed the other methods, including Cyclone and reCAT, which used known cell-cycle genes to optimize their performances. Further tests also showed that Cyclum performs better than NMF, ICA and PCA Supplementary Fig. 2g). We also performed gene set enrichment analysis (GSEA)[28,29] of the genes discovered ab initio using Cyclum, PC1, and PC2 on the mESC dataset. Cyclum yielded a normalized enrichment score (NES) for cell-cycle genes of 1.57, which compared favorably with the PC1 and PC2 scoring of 1.07 and 1.06, respectively, showing that Cyclum can better infer cell-cycle genes than the principal components.

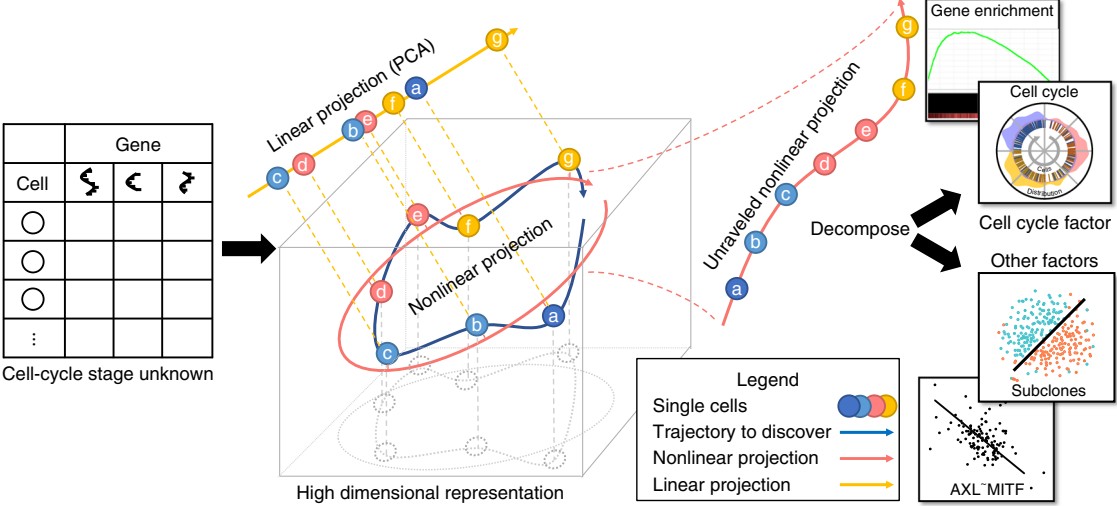

**Fig. 1 Overview of the Cyclum program.** Single-cell RNA-seq data in the format of a cell-gene expression matrix are given to Cyclum, which identifies a circular trajectory consisting of cells at different times (indicated by letters) and stages (labeled by colors) in the high-dimensional gene expression space (illustrated by a cube). Cyclum unravels the circular trajectory (red arrows) along with the projected cells to infer their pseudo-time. In contrast, a linear projection (yellow arrow) would result in incorrect ordering and timing. The inferred genes and pseudo-times can be further analyzed to discover new functions, cell-types, and cell-phenotype associations.

**Table 1 Labeled cell-cycle scRNA-seq datasets.**

| Dataset | Assaying | Labeling | Cell count | | | | Gene count |
|---|---|---|---|---|---|---|---|
| | | | Total | G0/G1 | S | G2/M | |
| mESC | scRNA | Hoechst | 288 | 96 | 96 | 96 | 38,293 |
| PC3 (CRL-1435) | qPCR | Hoechst | 361 | 85 | 141 | 135 | 253 |
| MDA-MB-231 (HTB-26) | qPCR | Hoechst | 342 | 123 | 103 | 116 | 253 |
| H9 (HTB-176) | qPCR | Hoechst | 227 | 66 | 68 | 93 | 253 |

We further assessed the robustness of Cyclum as related to sample size. We randomly subsampled the mESC data for fewer cells or genes. Stratified subsampling was used to keep an equal number of cells in each stage. Here, dimensionality of Cyclum is restricted to one to accelerate computing (see Methods), although it slightly reduces the accuracies. We observed that the median classification accuracy of Cyclum (ranging between 0.7 and 0.75) remained largely invariant with regard to the number of cells. In contrast, the median accuracy of reCAT became substantially worse with fewer cells (Fig. 2c). The variance increased with fewer cells for both programs. In a parallel experiment, we uniformly randomly subsampled genes. The accuracy of Cyclum was unaffected when there were over 10,000 genes (Fig. 2d). However, reCAT performed substantially worse with fewer genes and failed to return results when there were less than 5000 genes.

**Separability of subclones after corrected for cell cycle.** We assessed the utility of Cyclum in reducing the confounding effects introduced by cell cycle. A tissue sample often consists of multiple types of cells (e.g., tumor subclones) with distinct transcriptomic profiles[1,30]. When the cells are actively cycling, it can become difficult to delineate the cell types.

To assess the utility of Cyclum in this setting, we generated a virtual tumor sample consisting of two proliferating subclones of similar but different transcriptomic profiles. We used the mESC data as one clone and created a second clone by doubling the expression levels of a randomly selected set of genes containing variable numbers of known cell-cycle and non-cell-cycle genes

(see Methods). We then merged cells from these two clones together into a virtual tumor sample. This strategy allowed us to use real scRNA-seq data, although the perturbations applied are artificial. More importantly, it allowed us to track the clonal origins of each cell in the mixed population. We then ran Cyclum, ccRemover, Seurat, and PCA on the virtual tumor samples created under a wide range of parameters and assessed the accuracy of the algorithms in delineating cells from the two subclones. Cyclone and reCAT cannot remove cell-cycle effects, thus they were not included in the assessment.

We found that cells from the two subclones in a virtual tumor sample are intermingled in the t-SNE plot generated from the unprocessed scRNA-seq data (Fig. 3a). After removing cell-cycle effects using Cyclum, cells in the two subclones became separable (Fig. 3b). We then performed systematic assessment under a range of parameters, including the number of cells, number of perturbed genes, and the fraction of cell-cycle genes. We used a two-component Gaussian mixture model to quantify how well the two subclones were separated (classification accuracy) in the t-SNE plot. Under almost all conditions, Cyclum achieved significantly higher accuracy than the other methods, particularly when a large number (>400) of cell-cycle genes were perturbed (Fig. 3c and Supplementary Fig. 3). In contrast, approaches such as Seurat and ccRemover, which rely on the known cell-cycle genes, performed worse, especially when more cell-cycle genes were perturbed. These results demonstrated the benefit and robustness of Cyclum in deconvolving cell-cycle effects from the scRNA-seq data.

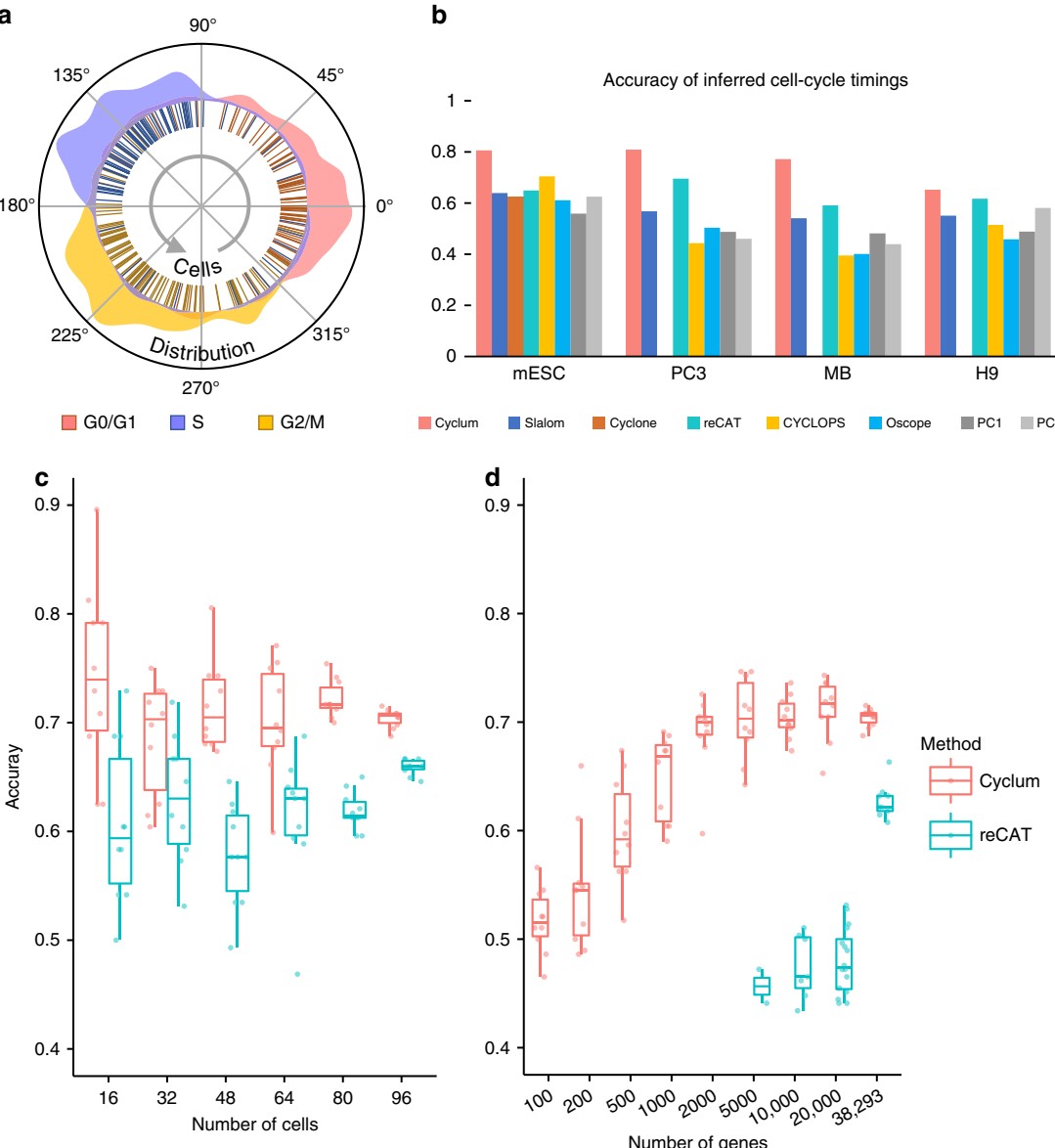

**Fig. 2 Accuracy of cell-cycle inference. a** Distribution of mESC cells over inferred cell-cycle pseudo-time. Curves outside are empirical distributions for cell-cycle stages and the lines inside represent individual cells. Colors correspond to ground-truth obtained experimentally by flow-sorting. **b** Cell cycle classification accuracy of Cyclum compared with Slalom, Cyclone, Cyclops, Oscope, reCAT, and principal component (PC) 1 and 2. **c** Cell-cycle classification accuracy (Y-axis) of Cyclum (red) and reCAT (blue) over the number of cells (X-axis) obtained from $n = 10$ randomly down-sampled datasets. **d** Cell-cycle classification accuracy (Y-axis) over the numbers of genes (X-axis, log-scale) obtained from $n = 10$ randomly down-sampled datasets for Cyclum, and $n = 30$ for reCAT. Only successful runs of reCAT were kept for plotting. For (**b**) and (**d**), the median (center lines), interquartile range (hinges), 1.5× interquartile range (whiskers), and corresponding data points (dots) are shown. Due to randomness in clustering, each method was evaluated five times on each dataset and the best accuracy is recorded. In this context, $n$ denotes sample size, not to be confused with number of cells in Methods. For all subpanels, source data are provided as a Source Data file.

**Application of Cyclum to the melanoma data.** We further examined the utility of Cyclum in analyzing scRNA-seq data obtained from real cancer samples. We examined the dataset consisting of the RNA expression of 23,686 genes in 4645 single cells from 19 melanoma patients, profiled using the 10X Chromium technology[31].

We analyzed the data from the five patients (i.e., Mel78, 79, 80, 81, and 88) that had over 100 cancer cells. First, we assessed how accurately Cyclum could characterize the cell cycle. We compared the pseudo-time inferred by Cyclum, reCAT, PC1, and PC2 against the GO:0007049 GO_CELL_CYCLE gene set using the GSEA. A higher GSEA score indicates that the pseudo-times inferred are more accurately tracing cell cycle. Cyclum performed

the best in this analysis (Fig. 4a and Supplementary Table 1), even on samples that reportedly had few cycling cells (e.g., Mel79). Among the novel cycling genes nominated (see Methods) by Cyclum (Supplementary Table 2 and Supplementary Figs. 4, 5), *KCNQ1OT1* and *FBLIM1* have been shown recently in the literature to be related to proliferation and tumorigenesis[32–37].

We estimated the proportions of cycling cells in these samples using Cyclum. Although Cyclum does not directly model quiescent cells, samples with fewer cycling cells (e.g., MEL79) appeared to have large gaps in the inferred pseudo-times (Supplementary Fig. 6). These gaps corresponded well to the missing S, G2, and M stages in these samples. In contrast, samples with a large fraction of cycling cells (e.g., the mESC) had largely

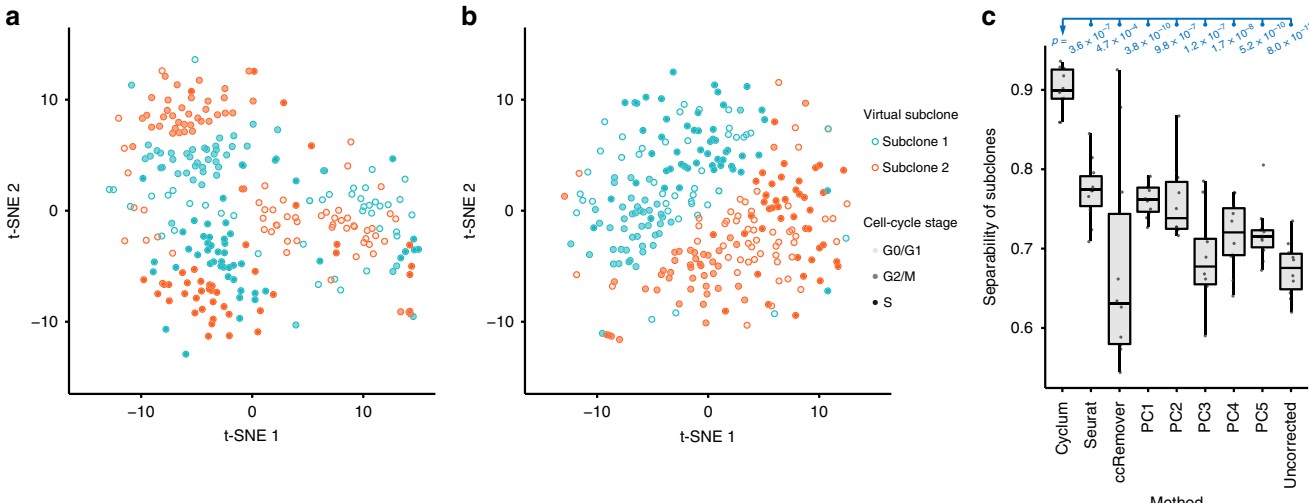

**Fig. 3 Subclone detection from virtual tumor data. a** t-SNE plot of the virtual tumor data consisting of two subclones (blue and red dots) of 288 cells each at various cell-cycling stages (shades). **b** t-SNE plot of the data corrected for cell-cycling effects using Cyclum. **c** The separability of subclones of $n = 10$ randomly generated virtual tumor datasets corrected by Cyclum, Seurat, ccRemover, principal component (PC) 1~5, and the uncorrected data. The median (center lines), interquartile range (hinges), 1.5× interquartile range (whiskers), and corresponding data points (dots) are shown. $P$-values were calculated using two-tailed Student's $t$-test. The expression levels of 1600 genes, including 600 known cell-cycle genes that were doubled in creating the virtual tumor data. Each method was evaluated five times on each dataset and the best accuracy is recorded. In this context, $n$ denotes sample size, not to be confused with number of cells in Methods. For all subpanels, source data are provided as a Source Data file.

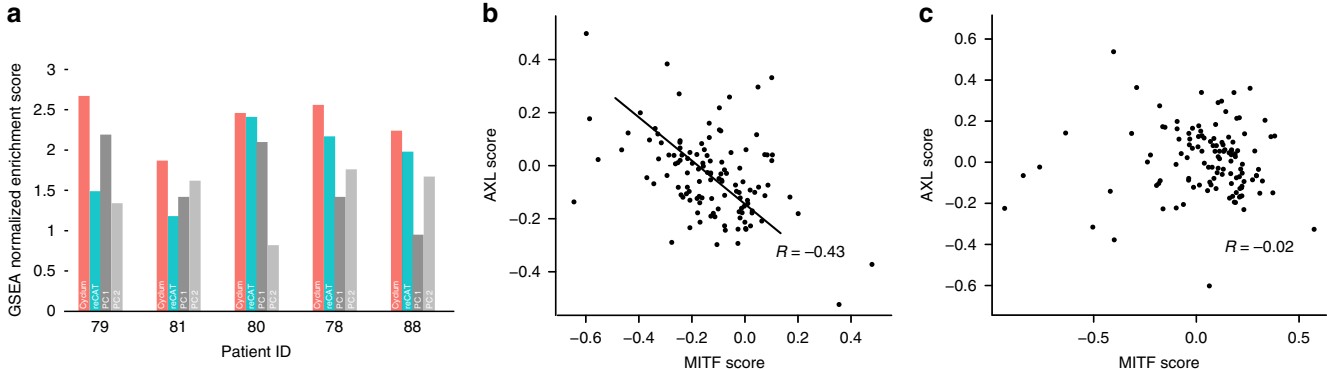

**Fig. 4 Cyclum results on the melanoma data. a** GSEA NES scores were obtained based on pseudo-times inferred by Cyclum, reCAT, and principal component 1 and 2. **b** The correlation between the MITF score and the AXL score for sample Mel78, based on Cyclum corrected expression data. **c** The correlation based on the uncorrected expression data. The AXL and MITF scores were calculated based on the average expression levels of the reported genes in AXL and the MITF program[31]. The lines in (**b**) were drawn manually for visual reference. The $R$ values are the Pearson correlation coefficients. For all subpanels, source data are provided as a Source Data file.

continuous pseudo-times. By partitioning the pseudo-time densities, we estimated the fraction of cycling cells in each sample (Supplementary Table 3). The resulting fractions appeared consistent, but were generally higher than those estimated based on the expressions of marker genes[31]. That could be expected, as Cyclum summarized signals from a larger set of genes showing periodicity.

Two dormant drug resistance programs (MITF-high and AXL-high) were present mutually exclusively in these melanoma patient samples, based on the immunofluorescence staining data[31]. To calculate the AXL/MITF program scores, we followed the method and gene sets suggested in the original publication[31]. The scores were defined as the average expression of the sets of genes. However, cell cycle could confound the expression profiles of these cells[19], making it difficult to delineate the resistance subgroups. Indeed, before correcting for cell-cycle effects, almost no correlation (Fig. 4c, $R = -0.02$, $P = 0.81$) was observed between the expressions of the cells from the two mutually

exclusive programs in the actively proliferating sample Mel78. After applying Cyclum correction, a clearly negative correlation (Fig. 4b, $R = -0.43$, $P = 9 \times 10^{-7}$) emerged, which is consistent with the expected mutual exclusivity between the two programs in single melanoma cells. The result was also better than that obtained using ccRemover and Seurat (Supplementary Fig. 7).

**Broader applications of Cyclum**. To evaluate Cyclum on larger high-throughput single-cell RNA-seq dataset, we applied it on a human embryonic stem cell (hESC) dataset containing 12,280 cells generated by the 10X Chromium technology, from a study of nicotine effects[38]. Results show that Cyclum accurately and efficiently infers cell-cycle pseudo-time (Supplementary Fig. 8a and Supplementary Note 2). Targets of nicotine, including *LDHA*[38] and *ENO1*[39] are proposed by evaluating the difference of circular pattern in each gene between the treated sample and the control sample (Supplementary Fig. 8b and Supplementary Note 2). To

further demonstrate application of Cyclum on processes other than cell cycle, we investigated the nonlinear epithelial-mesenchymal transition (EMT) and mesenchymal-epithelial transition (MET) on an mESC dataset[12,40]. Two transition points are identified and unique EMT and MET marker genes are extracted from them, respectively (Supplementary Fig. 9 and Supplementary Note 3). These results demonstrate Cyclum's broad utility over various biological questions and technologies and its scalability over large datasets.

## Discussion

In this work, we developed a trajectory inference method that can effectively characterize latent periodic developmental processes, such as cell cycle from scRNA-seq data. Compared with currently available methods, most of which are based on linear representations of the data, our approach can more effectively capture the nonlinearity and achieve more accurate characterization of a periodic process.

We examined Cyclum using multiple real and synthetic datasets. Using cancer cell lines and mouse embryonic stem cell data, we demonstrated that Cyclum accurately infers cell-cycle timing from the gene expression profiles of single cells, which was validated by flow-sorting results obtained independently on the same set of cells. Using virtual tumor data, we showed that Cyclum can be applied to remove confounding cell-cycle effects and achieving an improved classification of distinct cell subpopulations. Although the virtual tumor datasets cannot fully replace real data, the inputs were created under a wide range of parameters that facilitated systematic assessment of Cyclum and other comparable programs. Using the real datasets obtained from melanoma patients, we showed that Cyclum can accurately infer the cell-cycle expression components, nominate novel cell-cycle genes (e.g., *KCNQ1OT1* and *FBLIM1*), and elucidate latent associations between cell subpopulations and drug resistance. Cyclum also showed potential utility on other possibly circular processes such as epithelial-mesenchymal transition, endothelial-mesenchymal transition and their reverse processes[13,14]. These experiments indicated that Cyclum can be applied as a generic tool for characterizing periodic processes and discovering biologically meaningful cell subpopulations from scRNA-seq data.

We anticipate that Cyclum will be able to impact several important areas of investigation. First, it may be applied to discovering new genes involved in a periodic process, particularly genes that have a transitional or relatively low expression, and whose relevance is only evident when being observed across time (Supplementary Note 2). Second, it can be applied to remove cell-cycle effects and enhance the characterization of cell types and developmental trajectories. These utilities will be in great demand by the Human Cell Atlas[41], the Human Tumor Atlas Network, and many other projects.

It is worth noting that Cyclum is a model-based approach that fits the data to predefined circular manifolds. This design makes Cyclum more robust to handle random noise and small sample sizes. It constitutes a clear advantage over other model-free approaches, such as reCAT, for the purpose of characterizing cell cycles. Evidently, Cyclum's demonstrated robustness to a reduced number of cells and genes makes it desirable to analyze current scRNA-seq datasets, which often suffer from cell-specific dropouts and amplification bias[42]. Cyclum also appeared to work better on data that was heavily confounded by cell cycles. This is an important feature for studying cancer data, as many cancer cells have heightened cell-cycle activities[43,44]. On the other hand, when the latent process does not fit the circular manifold well, the method may bring limited benefit.

Our study clearly demonstrated the advantage of fitting scRNA-seq data to circular manifolds in a variety of settings. We plan to further explore how to use Cyclum in conjunction with other methods to deconvolving data generated by more complex, intertwined processes. For example, we plan to explore Gaussian process latent variable models (GPLVM) to track a generic periodic manifold that is not restricted to sinusoidal functions in the high-dimensional expression space. GPLVM has been applied previously to model linear trajectories[45], but also can be expanded to model periodic trajectories using specialized nonlinear kernels. We also plan to investigate the potential of applying Cyclum to characterize other periodic processes, such as circadian rhythms[15] from scRNA-seq data as well. A previous study has clearly demonstrated the potential of sorting biological samples based on circadian rhythms inferred from bulk gene expression data[6].

Our work also demonstrated that unsupervised machine-learning techniques, such as autoencoders, can be successfully applied to model latent periodic processes, with innovations on the network architecture and activation functions. The Cyclum package is efficiently implemented in Python using Keras with TensorFlow and has been comprehensively tested. For example, Cyclum can analyze an scRNA-seq dataset consisting of 12,280 cells 33,694 genes in 40 min on a laptop computer with a GTX 960M GPU and 2 GB graphic memory (detailed in Supplementary Note 2). Cyclum can be easily scaled up to bigger datasets in a high-performance computer cluster.

In summary, we developed Cyclum, a machine learning approach that can effectively and efficiently infer latent circular trajectories from scRNA-seq gene expression data. It can also be applied to removing confounding cell-cycle effects, improving the classification of cell subpopulations, and enhancing the discovery of functional gene subsets. These features make Cyclum useful to constructing the Human Cell Atlas, the Human Tumor Atlas, and other cell ontologies.

## Methods

**The Cyclum**. The objective of Cyclum is to infer pseudo-time/embedding $\mathbf{x}_n$ for cell $n$ from its transcriptome profile $\mathbf{y}_n$, a column vector containing the expression levels of $G$ genes. Linear methods, such as PCA, find a linear transformation $\mathbf{x}_n = \mathcal{F}(\mathbf{y}_n) = \mathbf{W}\mathbf{y}_n$ and an inverse linear transformation $\hat{\mathbf{y}}_n = \mathcal{F}^{-1}(\mathbf{x}_n) = \mathbf{W}^{\mathrm{T}}\mathbf{x}_n$, where $\mathcal{F}(\cdot)$ denotes the transformation in general sense and $\mathbf{W}$ denotes the specific linear transformation matrix, such that the total error $\sum_{n=1}^{N} \left\| \mathbf{y}_n - \hat{\mathbf{y}}_n \right\|^2$ is minimized[46]. Cyclum follows similar formulations, except that the transformation functions $\mathcal{F}^{-1}(\cdot)$ and $\mathcal{F}(\cdot)$ are nonlinear periodic functions, which makes Cyclum sensitive to circular trajectories (Fig. 1).

We use autoencoder[46], a machine learning approach to realize this nonlinear transformation (Supplementary Note 4). Specifically, we adopt an asymmetric autoencoder (Supplementary Fig. 1a). In the encoder, we use a standard multi-layer perceptron with hyperbolic tangent activation functions (Supplementary Fig. 10 and Supplementary Note 5). In the decoder, we use cosine and sine as the activation functions in the first layer, followed by a second layer performing linear transformations. These transformations can be represented mathematically as

$$
\begin{aligned}
\mathbf{x}_n = \begin{bmatrix} x_n^{(\mathrm{circular})} \\ \mathbf{x}_n^{(\mathrm{linear})} \end{bmatrix} &= \begin{bmatrix} \mathbf{W}_3^{(\mathrm{circular})} \tanh\left( \mathbf{W}_2^{(\mathrm{circular})} \tanh\left( \mathbf{W}_1^{(\mathrm{circular})} \mathbf{y}_n + \mathbf{b}_1 \right) + \mathbf{b}_2 \right) \\ \mathbf{W}^{(\mathrm{linear})} \mathbf{y}_n \end{bmatrix} \\
&\overset{\Delta}{=} \mathcal{F}(\mathbf{y}_n) \text{ and } \hat{\mathbf{y}}_n = \hat{\mathbf{y}}_n^{(\mathrm{circular})} + \hat{\mathbf{y}}_n^{(\mathrm{linear})} \\
&= \begin{bmatrix} \mathbf{V}^{(\mathrm{circular})} & \mathbf{V}^{(\mathrm{linear})} \end{bmatrix} \begin{bmatrix} \cos x_n^{(\mathrm{circular})} \\ \sin x_n^{(\mathrm{circular})} \\ \mathbf{x}_n^{(\mathrm{linear})} \end{bmatrix} = \mathbf{V}\mathbf{x}_n \overset{\Delta}{=} \mathcal{F}^{-1}(\mathbf{x}_n),
\end{aligned}
$$

where $\mathbf{W}$'s and $\mathbf{b}$'s are the weight matrices and translation vectors of the encoder, and $\mathbf{V}$ is the weight matrix of the decoder. The encoder part is useful when there are a large number of cells. Data from these cells can be divided into minibatches and sequentially loaded into the memory to train the parameters.

We use the least square error as the optimization target with L2 regularization, formally

$$\underset{\mathbf{W}_i, \mathbf{V}}{\operatorname{argmin}} \sum_{n=1}^{N} \left\| \mathbf{y}_n - \hat{\mathbf{y}}_n \right\|_2^2 + \sum_i \alpha_i \|\mathbf{W}_i\|_L^2 + \beta \|\mathbf{V}\|_L^2,$$

where $\mathbf{W}_i$ refers to all the $\mathbf{W}$'s presented above. The network is implemented using Keras with TensorFlow, which optimizes the parameters using gradient descent. We take the modulus of $x_n^{(\text{circular})}$ to confine its range to $[0, 2\pi]$ after the optimization.

**Choosing dimensionality for the embedding**. The dimensionality of embedding (and the number of added linear components) impacts the accuracy of Cyclum. We determined the dimension of the latent layers ($k$) by comparing the mean squared error (MSE) reconstruction loss of Cyclum (containing one circular dimension) with that of the principle component analysis (PCA) of the same $k$ dimensions. We declare a $k$ being the optimal $k^*$, if it leads to a large decrease of MSE. The dimensionality may be manually given to reduce computing time. An example is provided in Supplementary Note 1.

**Removing cell-cycle factor**. We assume that cell-cycle has an additive effect on the log-transformed expression. The $\hat{\mathbf{y}}_n^{(\text{circular})}$ is the estimated cell-cycle effect in $\mathbf{y}_n$ and can be removed through subtraction. We then perform t-SNE on the resulting expression levels. For comparison, we use principal components to remove cell-cycle factor by back-transferring the designated principal component to the expression space and subtracting it from the expression levels. Seurat uses a linear model to find the relationship between gene expression levels and the S and G2M scores it assigns to each cell. The residuals are the expression levels with the cell-cycle factor removed. ccRemover uses the PCA as the backend to iteratively remove all factors correlated with given cell-cycle genes.

**Predicting marker genes**. Using a standard trigonometric identity, the cell-cycle factor of a gene $g$ in cell $n$ can be reformulated as

$$\hat{y}_{ng} = V_{g,\cos} \cos x_n^{(\text{circular})} + V_{g,\sin} \sin x_n^{(\text{circular})}$$
$$= A_g \left[ \cos \varphi_g \cos x_n^{(\text{circular})} + \sin \varphi_g \sin x_n^{(\text{circular})} \right]$$
$$= A_g \cos \left( x_n^{(\text{circular})} - \varphi_g \right),$$

where $\varphi_g$ is the peak timing of a gene $g$ and $A_g$ is the magnitude of the peak, determined by $\mathbf{V}_g^{(\text{circular})} = [V_{g,\cos} \quad V_{g,\sin}]$, the $g$'th row of matrix $\mathbf{V}^{(\text{circular})}$. This is an alternative view of the decoder matrix $V$. It means that the decoder assigns pseudo-time to each cell and gives each gene a peak timing and a peak magnitude (Supplementary Fig. 1b–d). The weight $A_g$ indicates the prominence of the circular pattern in gene $g$. Cyclum ranks genes by $A_g$ and select those with high $A_g$ as the marker genes. It is a standardized measurement that can be used to compare the relevance of a gene to the circular process across datasets. An example is available in Supplementary Fig. 8 and Supplementary Note 2.

**Preprocessing**. We used log2 transformed Transcripts Per Million (TPM) in our experiment for scRNA-seq data (the mESC, the EMT and the melanoma data). For qPCR data (the cell lines) we used the reported normalized log counts. For droplet data (the hESC data), we used read counts normalized and log2 transformed data. One should expect only a slight difference across count normalization methods, as Cyclum examines overall circular patterns, instead of specific values. We also did not filter out any genes or cells, as Cyclum is robust against noise. Standardization was performed on each gene, adjusting the mean expression to 0 and standard deviation to 1, so that Cyclum equally considered all the genes. For data generated by the droplet technologies such as the 10X 3' scRNA-seq, we recommend filtering out low-quality cells, such as those with high levels of mitochondria gene expressions and with low read counts[38]. We do not recommend filtering out particular genes. Genes expressing stochastically are automatically ignored by Cyclum due to their poor fitting to the model.

**Utilizing prior knowledge**. Although Cyclum is a de novo approach, it may benefit from using known gene sets in some scenarios. By default, Cyclum considers all genes equal. To apply the prior knowledge, one can allocate more weights in the MSE to a subset of genes. For examples and detailed discussion, see Supplementary Fig. 2d–f and Supplementary Note 1.

**Simulating virtual tumor data**. To simulate the second clone in the virtual tumor data, we randomly selected a set of cells from the first clone (i.e., the mESC data). We then randomly selected a set of genes from a list of 892 known cell-cycle genes (Supplementary Data 1) and a set from other genes, including those that may be affected by, but are not directly affiliated with cell cycle. We then doubled the expression levels of the selected genes in the selected cells. We also varied the number of cells and genes to simulate data collected from a variety of conditions.

**Evaluating accuracy of timings and separability of subclones**. For Cyclone, which outputs categorical cell-cycle phases for each cell, the accuracy is defined as the ratio of cells that are correctly classified. For PCA and reCAT, which output numerical embeddings (pseudo-times), the score is the precision of a best three-component GMM classifier on the embeddings[47]. Because the cell cycle is a continuous process, cells residing between cell-cycle stages (border cases) exist. The accuracy scores, evaluated based on discrete cell-cycle stages obtained from flow-sorting experiments, may underrate the continuous pseudo-time. We further assessed the accuracy of the inferred cell-cycle pseudo-time using GSEA against the GO:0007049 GO_CELL_CYCLE gene set[28,29], treating pseudo-time as a continuous phenotype and reporting the normalized enrichment score (NES) as the accuracy.

The separability of subclones is defined as the precision of a best two-component GMM classifier on the t-SNE embedding of the data. Labels known from independent experiments (i.e. flow-sorting or simulation) are used to evaluate the classifiers.

**Reporting summary**. Further information on research design is available in the Nature Research Reporting Summary linked to this article.

## Data availability

No new data was generated in this study. All original datasets are accessible (E-MTAB-2805, https://doi.org/10.1371/journal.pcbi.1003696, GSE72056, GSE125416, and GSE87038) through the original publications[17,27,31,38,40]. The source data underlying Figs. 2a–d, 3a–c, 4a–c are provided as a Source Data file.

## Code availability

The open source implementation of Cyclum is available at https://github.com/KChen-lab/Cyclum under the MIT License.

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

## Acknowledgements

This project has been made possible in part by grant number 2018-182735 to K.C., Human Breast Cell Atlas Seed Network Grant (HCA3-0000000147) to N.N. and K.C. from the Chan Zuckerberg Initiative DAF, an advised fund of Silicon Valley Community Foundation, grant RP180248 to K.C. from Cancer Prevention & Research Institute of Texas, and the Cancer Center Support Grant P30 CA016672 to P.P. from the National Cancer Institute.

## Author contributions

S.L., F.W., J.H., and K.C. developed the Cyclum algorithm. S.L. implemented the code. S.L., F.W., J.H., and K.C. designed the experiments and helped analyze the results. All authors helped to write the paper. All authors have read and approved this paper.

## Competing interests

The authors declare no competing interests.
