## [Peer Review File · Nature Communications]

Reviewers' comments:

Reviewer #1 (Remarks to the Author):

Liang et al. proposed a novel method called Cyclum to infer the latent periodic process and remove the cell-cycle effects from complex single cell RNA sequencing datasets. Compared with current methods, Cyclum provided more accurate estimation of cell-cycle characterization. In both simulated and read tumor datasets, Cyclum successfully removes the effects of cell-cycle to help better depict the cellular atlases. However, I still have some concerns about the method.

1. The structure of AutoEncoder adapted by Cyclum is novel, but also is lack of detailed discussions. How about the dimensions of the latent layers? Do these dimensions have substantial effects on the accuracy? For the encoder parts, I noticed the author used an "old-fashion" non-linear activation function 'tanh'. As far as I know, this may have no specific relationship with periodicity, but could capture various kinds of non-linear structures. So how is it guaranteed to form an embedding related to the specific cell-cycle information but not some other kinds? And why do not adopt other modern activations, such as relu or selu?
2. Similar to the former question, Cyclum decomposed the embedding layers into the circular and linear parts, but how about the possible other non-linear structures in the dataset, such as cellular differentiation, tumor subtypes? Are these kinds of structures coupled with which parts of the embedding layer? Overall, the author may give more intuitive and convincing explanations about the features learned by each layer.
3. To predict the marker genes, the authors formalized the expression in a standard trigonometric identity. Are the distributions of numerical range of the magnitude A_g very similar or different for various kinds of datasets? In the case of various numerical scale of the magnitude, is it necessary or possible to give some uniform measure of importance across datasets, such as the statistical significance?
4. The accuracy of inferred cell-cycle timings is improved by Cyclum compared with other methods. But the value is still somewhat low, around 0.6-0.7, while other methods give 0.5-0.6. (Figure 1) The author may give deeper discussions about which factors make the inference so difficult. The Cyclum is a totally unsupervised method. Would using the known cell-cycle signatures, such as the signature genes, boost the accuracy or not?
5. The largest dataset tested in this manuscript has just 4,645 cells. Is the method applicable for large-scale datasets and especially the 10X datasets? In the preprocessing procedure, the author claimed no filtering steps for genes and cells, and performing standardization for all genes. In the 10X datasets, it is easy to capture various kinds of low-quality cells. And it is not natural to treat all genes as the equal importance. (Some genes may not be expressed in any cell or only expressed in less than 10 cells in the 10X datasets.) I think the author should give more appropriate application guidelines for their method.

Reviewer #2 (Remarks to the Author):

Liang et al. present Cyclum, a method that uses an autoencoder to infer latent periodic processes underlying single-cell expression data. Although I find the approach potentially interesting, the relationship with existing methods needs to be clearly explained. It is not clear how novel the proposed approach is, because at least four important previous methods are not mentioned. Additionally, I have some questions about the method itself.

Major Comments:

1. There are actually quite a few existing methods designed specifically to recover cell cycle ordering from single-cell data. Although the authors cite a few such methods, the omissions are significant: Oscope was published in Nature Methods in 2015, and is possibly the most visible such method. Another significant omission is Cyclops, an autoencoder (with very similar name to

Cyclum) that infers latent periodic processes, published in PNAS from Junhyong Kim's group. Additionally, the successor to scLVM, called f-scLVM, should be mentioned. Finally, classic matrix factorization approaches, including NMF and ICA, can infer cell cycle quite accurately (see Stein O'Brien et al., Cell Systems, 2019). The authors need to discuss this previous work, put their proposed approach in this context, and likely compare to the existing approaches (certainly Cyclops and Oscope).

2. Do the authors envision any use cases for this method other than inferring cell cycle ordering? Are there any other types of latent periodic processes that it could be used for? If cell cycle is the only goal, it seems that some sort of supervised or semi-supervised approach (such as f-scLVM takes) should do better. So many cell cycle-related genes are known that you could likely improve the results by leveraging these known markers.

Minor Comments:

1. What is a "convoluted biological process"? I have never heard this term, and the authors use it several times but do not define it.
2. The reverse of "convolve" is "deconvolve", not "deconvolute".
3. The term "gradient descending" is used instead of "gradient descent" in the methods section.

Responses to reviewers

We are grateful for the in-depth reviews and constructive suggestions. Accordingly, we have substantially revised our manuscript. We added two additional datasets (including 10X 3' scRNA-seq data) to illustrate the broader utilities of Cyclum, performed side-by-side comparison with two more methods suggested by Reviewer 2, and further clarified our methodology by constructing additional simulation studies. We adjusted the corresponding parts in our manuscript. We also provided answer to each specific question as follows.

1 REVIEWER 1

1.1 STRUCTURE OF THE ENCODER

1.1.1 Question

The structure of AutoEncoder adapted by Cyclum is novel, but also is lack of detailed discussions. How about the dimensions of the latent layers? Do these dimensions have substantial effects on the accuracy? For the encoder parts, I noticed the author used an “old-fashion” non-linear activation function ‘tanh’. As far as I know, this may have no specific relationship with periodicity, but could capture various kinds of non-linear structures. So how is it guaranteed to form an embedding related to the specific cell-cycle information but not some other kinds? And why do not adopt other modern activations, such as relu or selu?

1.1.2 Answer

We appreciate these excellent questions. We determined the dimensionality (k) of the latent layers using a systematic tuning approach. In this approach, we compared the reconstruction loss in terms of mean squared error (MSE) of Cyclum (one circular dimension included) with that of the principle component analysis (PCA) using the same k dimensions. We declare a k being the optimal k^* , if it leads to the largest decrease of MSE than any other k 's. As an example, we show that this approach chose $k^* = 2$ on the mESC dataset (Figure R1a) and indeed resulted in better embedding result (Figure R1b) than that from a different choice $k^* = 1$ (Figure R1c).

We now detailed this systematic tuning approach in the Methods and included Figure R1 as Figure S8 in Additional file 1. We also added Figure R1c to Fig. 2 to visualize the results of Cyclum, which we believe will help readers better understand the work.

The choice of different activation functions in the encoder did not affect the ability of our method in capturing the circular component because the periodicity is guaranteed by the explicit use of the sine and cosine functions in the decoder. For clarification, we performed a set of simulation experiments to compare the effects of using hyperbolic tangent (tanh), rectified linear unit (ReLU), and scaled exponential linear unit (SeLU) as activation functions in the encoder. We simulated random datasets of circular trajectories. We fixed the learning rate and used Glorot Normal (Xavier), He Normal, and LeCun Normal initializers for tanh, ReLU and SeLU, respectively, as each activation function relies on unique initializers to achieve its best performance. We noticed that tanh performs marginally better in more cases in terms of MSE (e.g., Figure R2), while the convergence rates of approaches are largely similar.

Since the sizes of these neural networks are relatively small, the differences in time consumption across activation functions are inconsequential. We added a section in the Additional file 1 to discuss this simulation study.

Figure R1: Selection and influence of the dimensionality.

a) Improvement in fitting (i.e., relative differences in MSE) of Cyclum, compared with PCA, under the same dimensionality (k). X-axis shows the number (k) of dimensions and Y-axis shows the relative improvement in fitting, where higher is better.

b, c) Circular embedding of Cyclum with dimensionality $k = 2$ (b) and 1 (c), with 1 circular dimension. Colors correspond to cell-cycle phases known from flow-sorting. Curves are empirical distributions for the stages and the lines represent individual cells. Accuracy evaluated against flow-sorting is 0.806 and 0.639, respectively.

Figure R2: Comparison of activation functions on simulated sinusoidal datasets. X-axis: number of training epochs. Y-axis: mean squared error loss of the model.

1.2 ABILITY OF HANDLING OTHER NONLINEAR COMPONENTS

1.2.1 Question

Similar to the former question, Cyclum decomposed the embedding layers into the circular and linear parts, but how about the possible other non-linear structures in the dataset, such as cellular differentiation, tumor subtypes? Are these kinds of structures coupled with which parts of the embedding layer? Overall, the author may give more intuitive and convincing explanations about the features learned by each layer.

1.2.2 Answer

We expect other nonlinear structure in the dataset to be either captured by the additional linear components, or left in the regression residual, as linear components can represent a nonlinear manifold with the cost of using multiple dimensions. Further visualization (e.g., UMAP) of those components will reveal other nonlinear structures. We simulated a dataset containing both a circular and an additional nonlinear component using both sinusoid and sigmoid functions. Gene expression levels in single cells are determined by a sinusoidal circular pattern, a sigmoidal jump pattern, or both (via summation). We expect that with the help of an additional linear component, Cyclum can still accurately recover the periodicity without being affected by the additional sigmoidal jump. If the model performed accurate inference, we would observe that the predicted pseudotime in the circular dimension well match the simulated time (manifesting as a diagonal line in the Fig. R3). The results we obtained largely matched our expectation, showing a diagonal curve in the circular time correspondence plot, with little displacement (Figure R3). The sigmoid pattern is partially captured by the linear component, and partially left in the residue. We added a section and figure S7 in the Additional File 1 to clarify this point.

Figure R3: Using a combination of linear and circular components to decipher combinational effects of a sigmoid process and a sinusoid process. X-axis: simulated time, the ground truth; Y-axis: inferred pseudotime in the circular dimension by Cyclum; both mod 2π . A diagonal pattern indicates good match of the prediction and the ground truth. A shift in phase (bottom right) in the predicted pseudotime is considered normal (i.e., the portion under $Y = 0.8$ may be cut and paste to the top of the figure to form a complete diagonal).

1.3 STANDARDIZED MEASUREMENT

1.3.1 Question

To predict the marker genes, the authors formalized the expression in a standard trigonometric identity. Are the distributions of numerical range of the magnitude A_g very similar or different for various kinds of datasets? In the case of various numerical scale of the magnitude, is it necessary or possible to give some uniform measure of importance across datasets, such as the statistical significance?

1.3.2 Answer

Good question. As a common practice in deep learning, the input for the neural network is standardized. Consequently, the magnitude A_g is a de facto standardized measurement and thus should be comparable across datasets. As a concrete proof, we tested our approach on a new 10X hESC dataset

(also to address the questions in section 1.5.1), GSE125416, which includes two samples, a control group of 6,766 cells and a nicotine treated group of 5,514 cells [1]. As is shown in Figure R4, we compared the inferred circular magnitude of genes A_g from the two datasets. The majority of genes lie on the diagonal, showing that the circular magnitudes of most genes are indeed comparable across datasets. We did notice a few exceptions. A_g of gene *LDHA* in the treated group increased the most. Interestingly, *LDHA* is a gene that regulates cell proliferation which is known to be downregulated post nicotine treatment [1]. The fact of A_g increased in the treated group shows that A_g is a measurement robust to the expression level changes, a nice property which may benefit biological discovery. The second gene that shows significantly stronger circular pattern in the treated dataset is *ENO1*, a known target of nicotine, which plays roles in cell proliferation and non-small cell lung cancer, of which tobacco usage is a leading risk [2]. In addition, the A_g of a set of mitochondrial genes are decreased in the treated group, which also conforms to a recent report conjecturing mitochondria as a target of nicotine [3].

Figure R4: Comparison of the magnitudes (A_g) across datasets. Each dot represents a gene, whose x-coordinate is its circular magnitude in the control group, and y-coordinate is that in the nicotine treated group.

We included part of the above discussion and figures into the revised manuscript (“Methods: Predicting marker genes” and Additional File 1: Figure S9 and Section 3.1.3) to demonstrate the generalizability and robustness of Cyclum on various datasets, technologies and experiments.

1.4 ACCURACY AND PRIOR KNOWLEDGE

1.4.1 Question

The accuracy of inferred cell-cycle timings is improved by Cyclum compared with other methods. But the value is still somewhat low, around 0.6-0.7, while other methods give 0.5-0.6. (Figure 1) The author may give deeper discussions about which factors make the inference so difficult. The Cyclum is a totally

unsupervised method. Would using the known cell-cycle signatures, such as the signature genes, boost the accuracy or not?

1.4.2 Answer

Thanks for the suggestions. Because the cell cycle is a continuous process, cells residing between cell-cycle stages (border cases) exist. The accuracy scores, evaluated based on discrete cell-cycle stages obtained from flow-sorting experiments, underrate the continuous pseudo-time. We have added this justification to our revised manuscript (“Methods: Evaluating accuracy of timings and separability of subclones”).

We also attempted to further improve the results by properly selecting the hidden layers (using our systematic tuning approach) and utilizing prior knowledge on cell-cycle genes, as suggested by the reviewer. As shown in Figure R5 below, while using only marker genes for Cyclum analysis does not improve the performance, allocating more weights on them while still considering other genes further increased the accuracy from 0.806 to 0.840 (a 4% improvement).

Figure R5: Utilizing marker genes as prior knowledge in Cyclum. The panels are for (a) no prior knowledge, (b) only keep marker genes, and (c) set more weights on the marker genes. Curves are empirical distributions for each phase and the vertical lines beneath represent individual cells. Accuracies are 0.806, 0.799 and 0.840, respectively.

In practice, the extent and credibility of prior knowledge in different biological processes vary a lot. For example, very few marker genes are known for the mesenchymal-epithelial transition [4], which may form a circular trajectory with epithelial-mesenchymal transition (see section 2.2.2 below). Higher noise level may also adversely affect the reliability of marker genes, making borrowing information from other genes more important.

We added description/discussion in the manuscript (“Methods: Utilizing prior knowledge” and Additional File 1: 3.1.2) showing that utilizing prior knowledge can potentially improve the result, and updated our software package to allow the user to weight features based on their knowledge. We also updated results in Fig. 2 in the revised manuscript due to improvements in the model fitting.

1.5 APPLICATION TO LARGER DATASETS

1.5.1 Question

The largest dataset tested in this manuscript has just 4,645 cells. Is the method applicable for large-scale datasets and especially the 10X datasets? In the preprocessing procedure, the author claimed no filtering steps for genes and cells, and performing standardization for all genes. In the 10X datasets, it is

easy to capture various kinds of low-quality cells. And it is not natural to treat all genes as the equal importance. (Some genes may not be expressed in any cell or only expressed in less than 10 cells in the 10X datasets.) I think the author should give more appropriate application guidelines for their method.

1.5.2 Answer

We have now tested our results on an additional 10X hESC dataset which includes 12,493 cells [1]. We followed the filtering method used in the original article and kept 12,280 cells, but only performed a regular scaling without regressing out any factors or filtering out any genes. As indicated by our automatic tuning approach, we used Cyclum with $k^* = 1$ (i.e., no additional linear component). Because the cells are not sorted by flow cytometry, we used GSEA to test if the genes inferred by Cyclum are cell cycle genes (see Methods in our revised manuscript for details.) We also tested Cyclops, reCAT, and Oscope [5] on this dataset (as suggested by Reviewer 2). Cyclum obtained the highest score (Table R1). Oscope does not scale up and were not able to return a result within 120 hours, while reCAT took 44 hours to finish. In contrary, Cyclum were able to return results in 40 minutes (i.e., 66 times as fast) on a personal computer equipped with a moderate NVidia GTX960M GPU. Furthermore, larger datasets will not significantly increase training time, as neural network benefits from stochastic gradient decent and early stopping [6].

Table R1: Comparison of GSEA score on hESC 10X dataset

	Cyclum	Cyclops	reCAT	Oscope	PC1	PC2	PC3	PC4
GSEA	2.82	1.22	2.50	NA	2.18	1.40	2.64	1.99
NES				(> 120h)				

We do suggest following the standard of single-cell data analysis workflow [1,7] to filtering out the low-quality cell, for example, the ones with abnormally high mitochondria gene expressions, low UMI or low read counts. As for genes, the experiments show that Cyclum is workable on the dataset without filtering. In our experience, the rarely expressed genes make statistically uncorrelated noise and did not appear to distract Cyclum from finding the correct embedding.

We have added these information to the revised manuscript (“Methods: Preprocessing” and Additional Files 1: Section 3.1.3) and online manual at <https://github.com/KChen-lab/Cyclum>.

2 REVIEWER 2

2.1 COMPARISON WITH OTHER METHODS

2.1.1 Question

There are actually quite a few existing methods designed specifically to recover cell cycle ordering from single-cell data. Although the authors cite a few such methods, the omissions are significant: Oscope was published in Nature Methods in 2015, and is possibly the most visible such method. Another significant omission is Cyclops, an autoencoder (with very similar name to Cyclum) that infers latent periodic processes, published in PNAS from Junhyong Kim’s group. Additionally, the successor to scLVM, called f-scLVM, should be mentioned. Finally, classic matrix factorization approaches, including NMF and ICA, can infer cell cycle quite accurately (see Stein O’Brien et al., Cell Systems, 2019). The authors need

to discuss this previous work, put their proposed approach in this context, and likely compare to the existing approaches (certainly Cyclops and Oscope).

2.1.2 Answer

We appreciate the suggestions. We were able to perform direct comparison with Oscope and Cyclops, which infer pseudotime. Several accommodations were made to run these methods on the datasets. The original focus of Cyclops is inferring circadian rhythm pseudotime. Although cell cycle is a topic in the report, it is indirectly modeled. Directly applying Cyclops on the cell cycle inference resulted in too few eigengenes due to its eigengene (PCA) filtering. To obtain the best results, we adjusted the threshold and kept the one yielding the best accuracy. Oscope was readily applicable to the mESC dataset. However, it did not find any set of genes as good indicator of the pseudotime on the PC3, MB and H9 datasets. To enable comparison, we manually recovered its candidate gene sets filtered out during the analysis. The overall result from all comparison experiments is summarized in Figure R6. Cyclum achieved evidently better accuracy than other methods.

Moreover, on the new 10X hESC dataset, which includes over 10,000 cells, Cyclum returned better result than reCAT and was much more time efficient (40 minutes versus 44 hours, i.e., 66 times as fast). Oscope was not able to return any result in 120 hours and Cyclops returned an inferior result (see section 1.5.2 for more details).

The f-sLVM (Slalom) and NMF approaches are both linear decomposing methods. Although they can similarly be applied to model cell-cycle processes, they do not directly calculate pseudotime and therefore cannot be included in side-by-side comparison. For example, for f-sLVM [8], although the predicted embedding of cells well separated cells in G2/M stage and G1 stage, cells in S stage were mixed with other stages (see Figure 2 in [8]). This makes it unsuitable to use the f-sLVM embedding as a cell-cycle pseudotime. Nonetheless, we cited f-sLVM and NMF approaches and discussed them in the “Background” of the revised manuscript, as suggested by the reviewer.

Figure R6: Performance of Cyclum versus other methods. Cyclone was not applicable on the PC3, MB, or H9 dataset.

We have revised our manuscript (Fig. 2) to include these updated results.

2.2 APPLICATIONS TO MORE BIOLOGICAL PROCESSES

2.2.1 Question

Do the authors envision any use cases for this method other than inferring cell cycle ordering? Are there any other types of latent periodic processes that it could be used for? If cell cycle is the only goal, it seems that some sort of supervised or semi-supervised approach (such as f-sLVM takes) should do better. So many cell cycle-related genes are known that you could likely improve the results by leveraging these known markers.

2.2.2 Answer

Yes. We envision that Cyclum will be used to study non-linear periodic processes other than cell cycle. As a demonstration, we performed a preliminary analysis on a mouse organogenesis dataset containing mRNA expression profiles of cells undergoing epithelial-mesenchymal transition (EMT). We were able to construct a circular trajectory (figure R7) [9], formed by EMT and its inverse process, the mesenchymal-epithelial-transition (MET), both of which are essential to organogenesis [10]. Unique MET marker genes (i.e., genes that only incur in MET but not EMT) such as *Cited1* and *Fzd7* [4], were identified in differential expression analysis when comparing the cells close to the two transition points. We have added these results into the revised manuscript (Additional File 1: Section 3.1.4, also referred to in the “Results” and “Discussions”).

Similarly, developmental processes like endothelial-mesenchymal transition (EndMT) and mesenchymal-endothelial transition (MEndoT), both occurring in cardiac neovascularization, may also form non-linear circular trajectories [11,12]. In addition, Cyclum may be used to study circular immune processes such as T cell exhaustion and reversal, or transitions between memory T cells and effector T cells [13,14].

We agree with the reviewer that utilizing known markers can improve the inference accuracy (see section 1.4.2). However, we would like to stress that although cell-cycle is relatively well-studied, there are rooms for further exploration. In that regard, Cyclum’s de novo approach can more effectively borrow information from non-marker genes to empower gene discovery or improve pseudotime inference accuracy. This ability is essential in scenarios with less reliable marker genes and noisier data.

Figure R7: Embedding of MET-EMT circular pseudotime in a preliminary experiment. Colors corresponds to the two cell types classified by the original study. Individual cells are arranged as bars inside the ring according to their pseudo-time. Curves outside the ring depict empirical density

distribution.

2.3 MINOR COMMENTS

2.3.1 Definition of *Convoluted Process*

Question: What is a “convoluted biological process”? I have never heard this term, and the authors use it several times but do not define it.

Answer: We have corrected “convoluted” to “convolved”, as it was brought to our attention (in section 2.3.2) that “convolved” is a more widely accepted term. We intended to express that multiple biological processes can occur concurrently and regulate gene expression jointly. We have rephrased it in the “Abstract” and “Background” for a better clarity.

2.3.2 Wording of *Deconvolve* and *Gradient Descent*

Suggestions: The reverse of “convolve” is “deconvolve”, not “deconvolute”. The term “gradient descending” is used instead of “gradient descent” in the “Methods” section.

Response: We are grateful for the meticulous reviews and have corrected our wordings as suggested.

3 REFERENCES

1. Guo H, Tian L, Zhang JZ, Kitani T, Paik DT, Lee WH, et al. Single-Cell RNA Sequencing of Human Embryonic Stem Cell Differentiation Delineates Adverse Effects of Nicotine on Embryonic Development. *Stem Cell Reports*. 2019;12:772–86.
2. Li MD. Tobacco Smoking Addiction: Epidemiology, Genetics, Mechanisms, and Treatment [Internet]. Springer Singapore; 2018 [cited 2019 Nov 9]. Available from: <https://www.springer.com/gp/book/9789811075292>
3. Malińska D, Więckowski MR, Michalska B, Drabik K, Prill M, Patalas-Krawczyk P, et al. Mitochondria as a possible target for nicotine action. *J Bioenerg Biomembr*. 2019;51:259–76.
4. Bult CJ, Blake JA, Smith CL, Kadin JA, Richardson JE, Anagnostopoulos A, et al. Mouse Genome Database (MGD) 2019. *Nucleic Acids Res*. 2019;47:D801–6.
5. Leng N, Chu L-F, Barry C, Li Y, Choi J, Li X, et al. Oscope identifies oscillatory genes in unsynchronized single-cell RNA-seq experiments. *Nat Methods*. 2015;12:947–50.
6. Lopez R, Regier J, Cole MB, Jordan MI, Yosef N. Deep generative modeling for single-cell transcriptomics. *Nat Methods*. 2018;15:1053–8.
7. Seurat - Guided Clustering Tutorial [Internet]. [cited 2019 Nov 16]. Available from: https://satijalab.org/seurat/v3.1/pbmc3k_tutorial.html

8. Buettner F, Pratanwanich N, McCarthy DJ, Marioni JC, Stegle O. f-sclVM: scalable and versatile factor analysis for single-cell RNA-seq. *Genome Biology*. 2017;18:212.
9. Dong J, Hu Y, Fan X, Wu X, Mao Y, Hu B, et al. Single-cell RNA-seq analysis unveils a prevalent epithelial/mesenchymal hybrid state during mouse organogenesis. *Genome Biology*. 2018;19:31.
10. Thiery JP, Acloque H, Huang RYJ, Nieto MA. Epithelial-Mesenchymal Transitions in Development and Disease. *Cell*. 2009;139:871–90.
11. Zeisberg EM, Tarnavski O, Zeisberg M, Dorfman AL, McMullen JR, Gustafsson E, et al. Endothelial-to-mesenchymal transition contributes to cardiac fibrosis. *Nat Med*. 2007;13:952–61.
12. Ubil E, Duan J, Pillai ICL, Rosa-Garrido M, Wu Y, Bargiacchi F, et al. Mesenchymal–endothelial transition contributes to cardiac neovascularization. *Nature*. 2014;514:585–90.
13. McKinstry KK, Strutt TM, Swain SL. The effector to memory transition of CD4 T cells. *Immunol Res*. 2007;40:114.
14. D’Cruz LM, Rubinstein MP, Goldrath AW. Surviving the crash: Transitioning from effector to memory CD8+ T cell. *Seminars in Immunology*. 2009;21:92–8.

Reviewers' comments:

Reviewer #1 (Remarks to the Author):

This revision addressed all the proposed issues. I recommend its publication for the journal.

Reviewer #2 (Remarks to the Author):

The authors have addressed some of my comments, expanding the discussion of related work to include key approaches and adding comparisons to Oscope and Cyclops. The authors showed that their method gave somewhat higher accuracies, higher GSEA score and better time efficiency. Additionally, I think that the addition of the analysis modeling EMT and MET as a circular process is somewhat interesting.

However, I still find that the description of related work is flawed, and the comparison with existing approaches is incomplete. In particular, scLVM is not based on a "linear representation generated from user-defined gene sets", but rather infers latent factors using a Gaussian Process Latent Variable Model (GPLVM), a nonlinear technique (which is, in fact, mentioned as a future direction in this paper). scLVM then identifies which of these factors corresponds to cell cycle by examining correlation with a set of cell cycle markers; the resulting factor gives a continuous cell cycle stage exactly analogous to that inferred here. Furthermore, the only scRNA-seq dataset with ground truth used for validation here (Buettner et al.) is from the scLVM paper and was gathered expressly for the purpose of validating scLVM's ability to identify a single latent factor corresponding to cell cycle stage! Therefore, I think that the authors need to include scLVM in their comparison. Additionally, the authors dismiss NMF and ICA because they are linear techniques, but I and others have found that such linear techniques are quite capable of identifying factors corresponding to continuous progression through the cell cycle. I therefore request that the authors also run ICA and NMF on their datasets, identify the factor with the highest correlation with the ground truth cell cycle stages, and calculate their metric on this factor. On a related note, PCA should similarly be evaluated based on the PC most correlated with cell cycle stage, rather than simply PC1 and PC2.

Minor Issues:

The first sentence of both the abstract and the background is still unclear. I don't understand what you mean that "biological processes convolve". Please elaborate, because this seems to be a key idea for motivating your approach, and you don't want to lose readers on the first sentence.

I also found some grammatical issues and non-idiomatic phrasing in the revised manuscript. For example, "after corrected for cell cycle" and "application of CycLum to the melanoma data".

Response to Reviewers

Thanks reviewer 2 for the insightful comments. We have carefully considered all the critiques and organized the questions and our responses into the following 3 sections.

1.1 COMPARISON WITH sclVM

1.1.1 Question

Reviewer: I still find that the description of related work is flawed, and the comparison with existing approaches is incomplete. In particular, sclVM is not based on a “linear representation generated from user-defined gene sets”, but rather infers latent factors using a Gaussian Process Latent Variable Model (GPLVM), a nonlinear technique (which is, in fact, mentioned as a future direction in this paper). sclVM then identifies which of these factors corresponds to cell cycle by examining correlation with a set of cell cycle markers; the resulting factor gives a continuous cell cycle stage exactly analogous to that inferred here. Furthermore, the only scRNA-seq dataset with ground truth used for validation here (Buettner et al.) is from the sclVM paper and was gathered expressly for the purpose of validating sclVM’s ability to identify a single latent factor corresponding to cell cycle stage! Therefore, I think that the authors need to include sclVM in their comparison.

1.1.2 Answer

This is a valid point. Our understanding of sclVM was derived from the following statement in the f-sclVM paper [1]: “our model is intrinsically linear and in particular it assumes that the inferred factors have linear additive effects on gene expression”. Perhaps that led to our slight misunderstanding and the exclusion of f-sclVM from our previous comparison.

We have now compared Cyclum with f-sclVM (also known as Slalom, a direct descendent of sclVM) using the cell-cycle dataset published in the sclVM paper, exactly as the reviewer suggested.

Figure RR1. A scatter plot of mESC cells projected onto the 2D space formed by the P53 pathway and G2M checkpoint factors inferred by Slalom. Each dot corresponds to a cell. The X-axis shows the loading over the G2M Checkpoint factor and the Y-axis over the P53 pathway factor.

We were able to reproduce the results in the original paper, thanks to the authors. The scatter plot generated using the top two Slalom factors (Figure RR1b) closely resembles the plot in the original

report (Fig 2a in [1]). Slightly different from the original report is that we included all the cells in the dataset (without filtering) to be fair to other methods in our assessment.

We estimated the classification accuracy of each Slalom factor by comparing inferred cell-cycle stages against the ground-truth labels obtained from flow-sorting. We converted the continuous cell cycle pseudotime into 3 discrete cell-cycle stages using a 3-component mixture Gaussian classifier (detailed in “Evaluating Accuracy of Timings and Separability of Subclones” in the Methods of our manuscript). Among all the Slalom factors, “G2M checkpoint” received the highest score 0.639, which is regarded as the accuracy of Slalom. The exact same processes were applied to the classification accuracies of all the methods in our assessment, to ensure a completely fair comparison. Slalom performed fairly well in this experiment. However, it is evident that Cyclum performed best across the 13 different methods (Figure RR2).

Figure RR2. Comparison of cell-cycle classification accuracies. Thirteen methods were compared using the mESC dataset.

For completeness, we also ran Slalom on the other smaller PC3, MB and H9 datasets, and reported the highest classification accuracies from the inferred factors.

We have now updated Fig. 2 in our revised manuscript to include the above results. We also adjusted the description of scLVM in the Introduction and the description of GPLVM in the Discussion of our revised manuscript, as suggested by the reviewer.

1.2 NMF, ICA, AND PCA

1.2.1 Question

Reviewer: Additionally, the authors dismiss NMF and ICA because they are linear techniques, but I and others have found that such linear techniques are quite capable of identifying factors corresponding to continuous progression through the cell cycle. I therefore request that the authors also run ICA and NMF on their datasets, identify the factor with the highest correlation with the ground truth cell cycle stages, and calculate their metric on this factor. On a related note, PCA should similarly be evaluated based on the PC most correlated with cell cycle stage, rather than simply PC1 and PC2.

1.2.2 Answer

We appreciate the suggestions on thoroughness. In response, we ran two additional methods, scCoGAPS (NMF) [2] and Sincell (ICA) [3], on the mESC datasets. These methods are published recently and have

been widely used/assessed. If multiple factors are detected, we chose the highest scoring one for pseudotime assessment. In addition to PC1 and PC2, we also extended comparison to PC3, PC4 and PC5. All these results are shown in Figure RR2 above.

We agree with the reviewer that linear methods are potentially very good at inferring some processes (e.g., G2M checkpoint) within the cell cycle. We argue that the nature and the extent of non-linearity throughout the entire cell cycle, which includes at least three major phases, is better represented by the sinusoid factors inferred by Cyclum, which is currently unavailable in the literature.

We have now included Figure RR2 as Figure S11 in the revised Additional file 1. We also briefly described those new results in the Results of our revised manuscript.

1.3 CLARITY

1.3.1 Question:

The first sentence of both the abstract and the background is still unclear. I don't understand what you mean that "biological processes convolve". Please elaborate, because this seems to be a key idea for motivating your approach, and you don't want to lose readers on the first sentence.

1.3.2 Answer

We have rewritten the first sentence.

We have also tried to reduce grammatical errors throughout the manuscript, in response to the reviewer's comments.

2 REFERENCES

1. Buettner F, Pratanwanich N, McCarthy DJ, Marioni JC, Stegle O. f-sclVM: scalable and versatile factor analysis for single-cell RNA-seq. *Genome Biology*. 2017;18:212.
2. Stein-O'Brien GL, Clark BS, Sherman T, Zibetti C, Hu Q, Sealfon R, et al. Decomposing Cell Identity for Transfer Learning across Cellular Measurements, Platforms, Tissues, and Species. *cells*. 2019;8:395-411.e8.
3. Juliá M, Telenti A, Rausell A. Sincell: an R/Bioconductor package for statistical assessment of cell-state hierarchies from single-cell RNA-seq. *Bioinformatics*. 2015;31:3380–2.

REVIEWERS' COMMENTS:

Reviewer #2 (Remarks to the Author):

The authors have satisfactorily addressed my comments. I have no further concerns.

No further question was raised by the referees. We would like to thank the referees for their suggestions.